# Effects of Four Weeks of Beta-Alanine Supplementation Combined with One Week of Creatine Loading on Physical and Cognitive Performance in Military Personnel

**DOI:** 10.3390/ijerph19137992

**Published:** 2022-06-29

**Authors:** Mohammad Samadi, Ali Askarian, Hossein Shirvani, Alireza Shamsoddini, Abolfazl Shakibaee, Scott C. Forbes, Mojtaba Kaviani

**Affiliations:** 1Exercise Physiology Research Center, Lifestyle Institute, Baqiyatallah University of Medical Sciences, Tehran 14155-6437, Iran; samadi.mohammad@yahoo.com (M.S.); a.askarian@ut.ac.ir (A.A.); shirvani.h2006@gmail.com (H.S.); shamseddin23@gmail.com (A.S.); shakibaee.abolfazl@yahoo.com (A.S.); 2Faculty of Education, Department of Physical Education Studies, Brandon University, Brandon, MB R7A 6A9, Canada; forbess@brandonu.ca; 3School of Nutrition and Dietetics, Faculty of Pure and Applied Science, Acadia University, Wolfville, NS B4P 2R6, Canada

**Keywords:** strength, anerobic performance, lactate, testosterone, cortisol, cognitive function

## Abstract

The purpose was to investigate the effects of a 7-day creatine (Cr) loading protocol at the end of four weeks of β-alanine supplementation (BA) on physical performance, blood lactate, cognitive performance, and resting hormonal concentrations compared to BA alone. Twenty male military personnel (age: 21.5 ± 1.5 yrs; height: 1.78 ± 0.05 m; body mass: 78.5 ± 7.0 kg; BMI: 23.7 ± 1.64 kg/m^2^) were recruited and randomized into two groups: BA + Cr or BA + placebo (PL). Participants in each group (n = 10 per group) were supplemented with 6.4 g/day of BA for 28 days. After the third week, the BA + Cr group participants were also supplemented with Cr (0.3 g/kg/day), while the BA + PL group ingested an isocaloric placebo for 7 days. Before and after supplementation, each participant performed a battery of physical and cognitive tests and provided a venous blood sample to determine resting testosterone, cortisol, and IGF-1. Furthermore, immediately after the last physical test, blood lactate was assessed. There was a significant improvement in physical performance and mathematical processing in the BA + Cr group over time (*p* < 0.05), while there was no change in the BA + PL group. Vertical jump performance and testosterone were significantly higher in the BA + Cr group compared to BA + PL. These results indicate that Cr loading during the final week of BA supplementation (28 days) enhanced muscular power and appears to be superior for muscular strength and cognitive performance compared to BA supplementation alone.

## 1. Introduction

Dietary supplements, including amino acids and a variety of other nutrients, are widely used by military personnel in the hopes of enhancing physical and cognitive performance [1]. Military personnel typically undergo strenuous physical training or operations that are often accompanied by physical fatigue and impaired cognitive function. Approximately 80–85% of military accidents are a result of diminished cognitive function [2]. For soldiers, the decline in physical and cognitive performance may elevate the risk of errors, which may ultimately result in a failed operation [3]. To counteract both physical and cognitive fatigue, military personnel often use dietary supplements [4]. Two of the most popular dietary supplements purported to enhance muscle and cognitive performance are creatine monohydrate (Cr) and beta-alanine [5].

Creatine, a nitrogen containing compound, is derived from three amino acids (arginine, glycine, and methionine) endogenously primarily in the liver and kidneys [6] and can also be synthesized in the brain [7]. It is well established that Cr supplementation elevates intramuscular phosphocreatine (PCr) and free Cr stores in skeletal muscle by ~20% [8] and ~5–10% in the brain [7]. This increases the capacity to re-phosphorylate adenosine triphosphate (ATP) during short-term intense physical exercise or during times of stress [9]. The progressive depletion of PCr during repeated intense activity increases the demands on the anaerobic glycolytic pathway, resulting in intramuscular accumulation of H^+^. Muscle acidosis impairs muscle function, including inhibition of phosphofructokinase (the rate limiting enzyme in glycolysis), impaired force production and contractility, and PCr resynthesis [10,11]. Therefore, Cr supplementation can facilitate greater energy capacity and reduce muscular acidosis, thereby enhancing physical performance. Further, PCr hydrolysis consumes H^+^ and therefore acts to buffer acidosis [12]. Cr also shuttles ATP from mitochondria to sites of ATP utilization, which may reduce oxidative stress [8]. In addition, creatine alters calcium handling which enhances myofibrillar cross-bridge formation and force production [8]. Overall, there is strong evidence that Cr supplementation can enhance exercise performance [8] and there is a growing body of literature showing beneficial effects on cognitive performance [7]. The precise mechanism(s) of how Cr supplementation alters cognitive function remains to be elucidated [7]. In theory, in the brain, Cr facilitates energy production and aids in maintaining energy status, improves mitochondrial efficiency, and reduces inflammation and oxidative stress, which may benefit brain function [7].

Another purported ergogenic aid and popular supplement is beta-alanine (BA). BA is a non-proteogenic amino acid and is the rate-limiting substrate in the synthesis of intramuscular carnosine [13] an intramuscular buffer [9]. A growing body of literature has shown that four to ten weeks of BA supplementation increases muscle carnosine concentration [9,14,15,16,17]. In addition, high-intensity physical exercise has been reported to benefit from the elevation of muscle carnosine [13]. Most of the scientific research investigating BA has focused on its role in muscle; however, an increase in brain carnosine in the cortex and hypothalamus has been observed using an animal model following daily BA supplementation [18]. In contrast, Hoffman et al., (2015) were unable to detect any changes in brain carnosine levels using magnetic resonance spectroscopy (MRS) in humans. Regardless, there is a positive correlation between brain carnosine and BDNF (brain-derived neurotrophic factor) [18,19]. Therefore, if brain carnosine could be elevated in humans following BA supplementation, it is possible that BA may increase cognitive function. 

Numerous studies have reported the advantages of Cr and BA supplementation independently on the performance of both athletes and military personnel [4,19,20,21,22,23]; however, much less research has examined the impact of co-ingestion. For example, there is evidence demonstrating a significant main effect of time following co-ingestion on several (5 out of 8) cardiorespiratory parameters including VO_2_ and power output at the lactate and ventilatory thresholds and %VO_2_ peak at the ventilatory threshold, while Cr alone only showed improvement in time to exhaustion and power output at the ventilatory threshold, and BA supplementation only improved power output at the lactate threshold [24]. Further, there is evidence that co-ingestion enhanced a single bout of anaerobic exercise [20] compared to placebo and Cr alone, but no comparison was made with BA. Furthermore, Okudan et al., (2015) found a significant improvement in mean power with co-ingestion during repeated anaerobic exercise in sedentary men [25]. In contrast, Kresta et al., (2014) did not report any additive benefits of BA and Cr supplementation compared to the placebo in recreationally active females [26]. 

To the best of our knowledge, no study has investigated the effects of BA and Cr co-ingested on a battery of physical tests with an assessment of cognitive function in military personnel compared to BA alone. Therefore, the purpose was to investigate the effects of Cr and BA supplementation on strength, anaerobic power, muscular power, blood lactate, and cognitive function compared to BA alone. A secondary purpose was to explore resting hormonal concentrations (i.e., IGF-1, testosterone, and cortisol), which may alter the anabolic/catabolic environment. Creatine supplementation has been shown to alter IGF-1 [27] and testosterone [20]; however, the effects of BA and Cr co-ingested on these hormones remains to be fully elucidated. 

## 2. Materials and Methods

### 2.1. Participants

Twenty recreationally active male soldiers (land forces in operation) between the ages of 19 and 25 years were recruited for this study. Participants were informed of all procedures, risks, and benefits and they signed an informed consent form. Participants were not permitted to use any additional nutritional supplements, anabolic steroids, or any other anabolic agents known to increase physical or cognitive performance. Exclusion criteria included any known kidney or liver disease, lower limb injury, and use of nutritional supplements in the preceding four weeks. In addition, all participants were omnivores. In the first session, participants completed a medical history questionnaire, and a physical activity readiness questionnaire (PARQ+). Informed consent was obtained from all participants involved in the study. The study was conducted in accordance with the Declaration of Helsinki and approved by the ethics committee at the Baqiyatallah University of Medical Science (IR.BMSU.REC.1399.593). 

### 2.2. Supplementation

Participants were randomly assigned into two groups (n = 10 per group). Both groups were instructed to ingest 6.4 g of BA per day (beta-alanine, pnc, 8 capsules, 800 mg each) for four weeks. In the last week, Cr (creatine Monohydrate, pnc, 0.3 g/kg per day) was added to the supplementation strategy for the BA + Cr group, and the other group (BA + PL) consumed an isocaloric isovolumetric placebo (equal grams of rice flour). Cr and PL were weighed according to each participant’s body mass and packaged in seven small bags for seven different days. Participants were asked to divide the powder into three equal portions, mix with water, and ingest it with their meals. Participants were instructed to split eight capsules of β-alanine (3 with breakfast, 3 with lunch and 2 with dinner) and consume the supplement with their regular meals. In addition, to avoid diet-induced changes in performance, all participants were asked to complete a 24 h food log on the day before the first testing trial and repeat the same diet the day before the second trial. In addition, participants were instructed to avoid intense exercise, alcohol and caffeine on the day preceding each experimental trial.

### 2.3. Testing Procedures

All participants performed a familiarization trial, and two main experimental testing sessions (baseline testing and post supplementation). Baseline and post testing were separated by a 28-day period of supplementation. Prior to the testing protocols, participants performed a 10 min warm-up that included a general warm-up and a specific warm-up. Following the warm-up, participants were able to rest (but the time did not exceed 3 min). Furthermore, the overall time that each participant spent in the laboratory was between 40 and 50 min. Therefore, we were able to test five participants each day between 11 a.m. and 3 p.m. Participants completed a battery of physical tests, which were completed in the following order. Each test was separated by a rest period which did not exceed 3 min. 

#### 2.3.1. Running Anaerobic Sprint Test (RAST)

The RAST evaluates anaerobic power and was performed as previously described [28]. Briefly, participants performed the RAST in six 35 m maximal sprints with a 10 s interval between each sprint. The time for each run was measured by two examiners, and the start for each sprint was announced with a whistle from the third examiner.

#### 2.3.2. One-Repetition Maximum Test

The one-repetition maximum (1RM) test is considered the gold standard to assess muscle strength. According to previously described protocol [29], bench press and leg press were performed to measure upper-body and lower-body strength, respectively. 

#### 2.3.3. Vertical Jump Test 

The vertical jump test is an indicator of leg muscle power. Briefly, participants stood with a wall at their right side and reached their right hand as high as possible while keeping their feet flat on the ground in order to mark their standing reach height. This standing reach height was also used in the post test. Then, they were asked to bend their knees, pause, and jump as high as possible and touch the wall with their finger at the peak of their jump, which was marked with tape. Each individual performed three jumps, and a vertical jump score was determined by subtracting the difference between the jump height and their standing reach height from their best attempt [30].

#### 2.3.4. Simulated Casualty Evacuation Test (SCET)

The SCET consists of a 1600 m run/walk, a 400 m walk where participants carry a rifle-shaped object (weight = 4 kg), and a 140 m casualty carry (weight = 70 kg). The weight was distributed through a 25 kg ruck sack on the participant’s back and one 22.5 kg dumbbell in each hand. The dumbbell carry was used to simulate a casualty carry. Prior to testing, the ruck sack was placed on each participant and the fit was adjusted properly. The participants were then instructed to bend down and pick up the two dumbbells that were placed on the floor at the starting line. Participants began walking to a cone placed 20 m away as fast as possible. Once the participant reached the cone, they were instructed to go around the cone and return to the starting cone seven times for a total distance of 140 m. Verbal encouragement was provided throughout the test. The time for each participant to complete the casualty carry was recorded. SCET was performed in an indoor, multipurpose court with a military uniform. The overall time of this test was recorded. 

#### 2.3.5. Mathematical Processing

A modified version of the original Serial Sevens test was used to assess cognitive function. Each participant was provided with a sheet of paper and a list of subtraction equations. Participants were instructed to subtract the number seven from each random four-digit number to measure how quickly and accurately they could calculate a simple mathematical equation in two minutes. The number of correct answers was recorded [4]. 

### 2.4. Biochemical Analysis

Blood samples were taken from the forearm vein following a 12 h fast at 8:00 a.m. 24 h before supplementation and 24 h after the last physical test. Serum concentrations of testosterone (Elecsys testostosteron II, Cobas, Mannheim, Germany, Cat. No: 05200067) and cortisol (Elecsys Cortisol II, Cobas, Mannheim, Germany, Cat. No: 06687733) were analyzed using a CLIA method (Cobas e411, Roche, Mannheim, Germany) and IGF-1 was analyzed via commercially available enzyme-linked immunosorbent assay (Mediagnost, Cat. No, E20, Mannheim, Germany). At the end of the last physical test (SCET), both in pre-test and post-test, blood lactate was immediately measured from the tip of index finger using a lactate analyzer (Lactate Scout (LS, SensLab GmbH, Mannheim, Germany)).

### 2.5. Statistical Analysis

Independent sample *t*-tests were used to analyze between group differences at baseline. The independent variables were then analyzed using Analysis of co-variance (ANCOVA; controlling for baseline values) to detect differences between the two groups over time. In addition, paired-sample *t*-tests were used to examine within-group changes before and after supplementation. Significance was set at *p* ≤ 0.05. All statistical analyses were performed using SPSS version 23.

## 3. Results

### 3.1. Participant Characteristics

Table 1 shows the demographic profiles of the participants in both groups. No significant differences in any of the participant demographic characteristics (age, height, body mass, and BMI) were observed between groups at baseline (*p* > 0.05). 

### 3.2. Physical and Cognitive Tests

The values of the physical tests including RAST, leg press, chest press, vertical jump, and SCET, as well as the mathematical processing scores, are shown in Table 2. Independent sample *t*-test showed no significant differences in these values at baseline (*p* ≥ 0.05). 

#### 3.2.1. Repeated Anaerobic Sprint Test (RAST)

No significant differences were observed in peak power (*F*_1,17_ = 0.131, *p* = 0.722), average power (*F*_1,17_ = 2.39, *p* = 0.140), minimum power (*F*_1,17_ = 2.32, *p* = 0.145), and fatigue index (*F*_1,17_ = 0.404, *p* = 0.533) between the two groups. When exploring the within-group comparison, there was a significant change in peak power (*p* = 0.014), minimum power (*p* = 0.031), and average power (*p* = 0.023) only in the BA + Cr group.

#### 3.2.2. Chest Press

No significant differences were observed between BA + Cr and BA + PL in 1RM of the chest press (*F*_1,17_ = 1.614, *p* = 0.224). For within-group comparisons, a significant increase of 1RM in BA + Cr (*p* = 0.026) and a non-significant change for the BA + PL group (*p* = 0.094) were observed.

#### 3.2.3. Leg Press

No significant differences were observed between BA + Cr and BA + PL for 1RM leg press (*F*_1,17_ = 0.266, *p* = 0.600). For within-group comparisons, a significant increase of 1RM in BA + Cr (*p* = 0.017) and a non-significant change for the BA + PL group (*p* = 0.071) were observed. 

#### 3.2.4. Vertical Jump

A significant difference was observed between BA + Cr and BA + PL in the vertical jump test (*F*_1,17_ = 10.53, *p* = 0.005). For within-group comparisons, a significant increase in the BA + Cr (*p* = 0.009) group and a non-significant change for the BA + PL group (*p* = 0.521) were observed.

#### 3.2.5. Simulated Casualty Evacuation Test (SCET)

No significant differences were observed between BA + Cr and BA + PL in SCET (*F*_1,17_ = 0.063, *p* = 0.600). Within-group comparisons showed a significant decrease in the overall time to complete the test in the BA + Cr (*p* = 0.003) group and the BA + PL group (*p* = 0.000). 

#### 3.2.6. 7-Mathematical Processing

No significant differences were observed between BA + Cr and BA + PL in the number of correct answers of this test (*F*_1,17_ = 0.045, *p* = 0.835). Within-group comparisons showed a significant improvement in the overall number of correct answers of the test in the BA + Cr group (*p* = 0.015), and a non-significant change for the BA + PL group (*p* = 0.138) was observed.

### 3.3. Blood Analyses

Values of all blood markers are presented in Table 3. There was no significant difference in these blood markers at baseline (*p* > 0.05).

#### 3.3.1. Testosterone

Changes in circulating testosterone concentrations are provided in Table 3. ANCOVA showed a significant difference between the BA + Cr and BA + PL groups (*F*_1,17_ = 9.73, *p* = 0.006). Moreover, within-group comparisons showed a significant increase of testosterone levels in the BA + Cr group (*p* = 0.001) and a non-significant change in the BA + PL group (*p* = 0.588). 

#### 3.3.2. Cortisol

Changes in serum cortisol concentrations are provided in Table 3. ANCOVA showed no significant difference between the BA + Cr and BA + PL groups (*F*_1,17_ = 0.0369, *p* = 0.551). In addition, within-group comparisons did not show any significant changes of cortisol level in BA + Cr (*p* = 0.481) and BA + PL (*p* = 0.981). 

#### 3.3.3. IGF-1

Changes in serum IGF-1 concentrations are provided in Table 3. ANCOVA showed no significant difference between the BA + Cr and BA + PL groups (*F*_1,17_ = 2.74, *p* = 0.116). Within-group comparisons did not show any significant difference in both the BA + Cr (*p* = 0.178) and BA + PL (*p* = 0.631) groups.

#### 3.3.4. Post-Exercise Lactate

Changes in lactate concentrations are shown in Table 3. ANCOVA did not show any significant differences between the BA + Cr and BA + PL groups (*F*_1,17_ = 3.22, *p* = 0.090). Within-group comparisons showed a significant decrease in blood lactate in both the BA + Cr (*p* = 0.002) and BA + PL groups (*p* = 0.045).

## 4. Discussion

The efficacy of BA and Cr supplementation individually are well documented to augment physical performance [20,26,31]; however, there is limited mixed evidence evaluating the effects of BA co-ingested with Cr. Furthermore, both Cr and BA may alter cognitive function. The present manuscript is the first study to investigate the effects of BA supplementation combined with Cr loading on both physical and cognitive performance in trained military personnel. The main findings revealed that BA + Cr further augmented vertical jump performance and elevated resting testosterone compared to BA + PL. Despite no interaction effect, we performed exploratory within-group comparisons which revealed significant improvements in RAST, chest press and leg press 1-RM, and the cognitive task in the BA + Cr group, while there were no differences over time for the BA + PL group. Both the BA + PL and BA + Cr groups improved SCET performance.

Cognition is complex and highly variable, which may be influenced by Cr [7]. There are a limited number of studies which have examined the effects of Cr on cognition, but there is evidence that Cr can improve short-term memory and intelligence/reasoning of healthy individuals by increasing the brain’s energy supply and its neuroprotective function, but its effect on other cognitive domains remains unclear [7,32]. In the present study, there was a significant improvement in processing speed when BA was combined with Cr, while there was no change in the BA alone group. These results support previous research, which has shown improvements in cognitive performance following Cr supplementation [7]. However, caution is warranted since there was no interaction effect, and thus, the analyses were exploratory in nature. Future large randomized controlled trials are needed to confirm these findings and to evaluate other cognitive domains.

It is reported that β-alanine supplementation results in higher physical work capacity, time to exhaustion and an increase in lactate threshold in both untrained males and females [33,34]. Β-alanine can increase intramuscular carnosine. Higher muscle carnosine concentrations have long been an important molecule to improve high-intensity non-oxidative exercise performance [13]. This is primarily due to carnosine’s function to enhance muscle buffering capacity during maximal and sub-maximal exercises [4,31,34]. In support of previous research, the present study demonstrated that BA significantly improved SCET time and reduced blood lactate compared to baseline; however, it is important to note that there was no placebo control group. Therefore, these results should be interpreted with caution. Cr supplementation has also been shown to have performance-enhancing properties due to several factors including increasing skeletal muscle phosphocreatine (PCr) content, improved PCr resynthesis, increased buffering capacity, and greater shuttling of mitochondrial ATP into the cytoplasm [8,24]. Although Cr loading plus BA significantly increased vertical jump performance, there were no significant changes in peak power and average power, as well as the time of completion of the SCET. In contrast, Zoller et al., (2007) reported beneficial effects of co-ingestion on several indices of cardiorespiratory fitness [24]. These differences can be partially attributed to the duration of the study, supplementation dose (28 days, 5.25 g Cr and 1.6 g BA), and testing protocols used. In the present study, the RAST protocol was the first physical assessment; since the duration of each bout was less than 6 s, it is not likely to stress the anaerobic systems capacity to significantly elevate H^+^ concentrations. In addition, most studies have not reported significant increases in maximal strength and power or muscular endurance (lasting less than 60 sec in duration) after BA supplementation [35,36,37]. Taken together, it appears that BA supplementation would be more effective in longer intense activities that stress the anaerobic glycolytic system, and Cr would be more beneficial to activities that require explosive power which are both in agreement with the findings of this study.

Furthermore, we hypothesized that Cr would enhance both upper and lower body strength. In support of this hypothesis, the within-group comparison showed a significant increase in chest and leg strength in the BA + Cr group, while no change was found in the BA + PL group. Several mechanisms exist by which creatine may increase lean tissue mass and muscular strength [38]. In several systematic reviews, Chillibeck et al., (2017), Lanher et al., (2015, 2016), and Forbes et al., (2021) reported that Cr supplementation is effective at improving upper and lower body strength [38,39,40,41]. Furthermore, in the present study, the BA + Cr significantly increased testosterone. Previous studies have demonstrated a positive correlation between testosterone and lean body mass and muscular strength [42,43,44]. Our results support previous research that reported a significant elevation in resting testosterone concentration [45,46] after Cr loading and no alteration following BA supplementation [21]. In contrast, some studies have shown that Cr loading did not change hormonal status [47,48]. Furthermore, the implications of a small rise in endogenous levels of testosterone on muscle adaptations are controversial and may only play a minor role [49].

Alterations in testosterone and cortisol concentrations are often used as indicators of the anabolic and catabolic environment, respectively, and have previously been used to quantify stress associated with prolonged military tasks [50,51,52]. Our data did not show any significant changes of cortisol concentrations between and within groups following supplementation. In contrast to the results of the current study, Varanoske et al., (2018) reported that, following BA supplementation, cortisol was significantly lower than PL group at 12 h after military tasks. In addition, they found a greater concentration of cortisol at 24 h compared to 0H after tasks [4]. As such, different time points used for sampling may explain these differences.

Between and within-group comparisons in our study indicated that BA and Cr supplementation elicited no significant change in blood IGF-1. There is limited evidence regarding BA and Cr supplementation on circulating IGF-1. Burke et al., (2008) reported that heavy resistance training increased muscle IGF-1 and supplementation with Cr resulted in greater increase [27]. This difference could be attributed to the duration of Cr supplementation and the exercise protocol.

There are several limitations to the current study that might have affected the results. The most important is that we were unable to take blood samples at different time points to see the response of the variables after physical activities. In addition, we were not able to measure creatine content in muscles to ensure that they were fully saturated. Finally, we did not have a placebo control group or a Cr-only group, which limited our abilities to assess the effectiveness of BA alone or to evaluate the synergistic effects of co-supplementation.

## 5. Conclusions

To the best of our knowledge, this is the first study examining the effects of one week of creatine loading in conjunction with four weeks of BA supplementation on physical and cognitive performance in military personnel. Four weeks of BA supplementation combined with 1 week of Cr loading significantly increased vertical jump (leg power) and resting testosterone compared to BA alone. Within-group comparisons revealed greater physical and cognitive performance when BA was supplemented with Cr compared to BA alone. Considering that the amount of physical and cognitive stress on army forces while maintaining health and performance is crucial, it would be deemed necessary to utilize effective nutritional strategies and dietary supplements to support their overall health and well-being as well as minimizing likelihood of injuries in any form. Further well-controlled studies are required to confirm these findings and to assess other cognitive domains.

## Figures and Tables

**Table 1 ijerph-19-07992-t001:** Demographics and anthropometrics of participants.

Variables	BA + Cr		BA + PL	
	Pre	Post	Pre	Post
Age (year)	21.4 ± 2.05		21.6 ± 2.01	
Body mass (kg)	77.2 ± 6.47	77.2 ± 6.71	74.5 ± 7.6	74.6 ± 6.8
Height (m)	1.79 ± 0.06	--------	1.77 ± 0.04	--------
BMI (kg/m^2^)	23.88 ± 1.22	23.88 ± 1.29	23.52 ± 2.04	23.55 ± 1.80

Values are presented as mean ± standard deviation. BA = β-alanine; Cr = Creatine; BMI = Body Mass Index.

**Table 2 ijerph-19-07992-t002:** Changes in physical and cognitive performance before and after 4 weeks of supplementation.

Variables	BA + Cr		BA + PL		*p*-Value
	Pre	Post	Pre	Post	
RAST PP (W)	539.48 ± 100.46	5.65.54 ± 104.28	522.43 ± 102.57	542.73 ± 113.64	0.722
RAST AP (W)	486.81 ± 87.03	499.26 ± 80.03	472.7 ± 90.38	475.89 ± 95.76	0.140
RAST MP (W)	423.2 ± 72.20	441.93 ± 66.9	408.90 ± 90.18	414 ± 88.66	0.145
RAST FI (W/s)	3.31 ± 1.19	3.55 ± 1.30	3.24 ± 1.66	3.68 ± 1.60	0.533
Chest Press (kg)	69.16 ± 8.15	71.06 ± 8.75	67.6 ± 9.54	68.4 ± 9.48	0.224
Leg Press (kg)	202 ± 16.4	205 ±17.71	201.5 ± 20.4	204.16 ± 19.47	0.600
Vertical Jump (cm)	53 ± 5.68	55.03 ± 8.29	55.86 ± 6.04	55.7 ± 6.03	0.005 *
SCET (min)	10.98 ± 0.53	10.32 ± 0.46	11.04 ± 0.62	10.37 ± 0.49	0.600
7-Subtraction Test (arbitrary unit)	9.30 ± 1.49	9.8 ± 1.22	9.5 ± 1.58	10 ± 1.24	0.835

Values are presented as mean ± standard deviation. RAST = repeated anaerobic sprint test; PP = peak power, AP = Average Power, MP = minimum Power, FI = Fatigue Index, SCET = simulated Casualty Evacuation Test. * *p* ≤ 0.05 show significant change between groups before and after 4 weeks of supplementation.

**Table 3 ijerph-19-07992-t003:** Resting hormonal levels and post-exercise blood lactate before and after 4 weeks of supplementation.

Variables	BA + Cr		BA + PL		*p*-Value
	Pre	Post	Pre	Post	
Testosterone (ng/L)	5.46 ± 0.66	5.92 ± 0.65	5.61 ± 0.59	5.64 ± 0.64	0.006 *
Cortisol (ng/L)	127 ± 7.7	126.4 ± 5.8	127.9 ± 8.5	127.8 ± 7.8	0.551
IGF-1 (pg/L)	280.5 ± 37.02	285.5 ± 32.35	281 ± 36.34	279 ± 28.46	0.116
Lactate (mmol/L)	12.98 ± 0.81	11.86 ± 0.73	12.84 ± 1.07	12.28 ± 0.65	0.090

Values are presented as mean ± standard deviation. * *p* ≤ 0.05 show significant changes between groups before and after 4 weeks of supplementation.

## Data Availability

The data presented in this study are available on request from the corresponding author.

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
