# Peer review of "Effects of Four Weeks of Beta-Alanine Supplementation Combined with One Week of Creatine Loading on Physical and Cognitive Performance in Military Personnel"

_ijerph, 2022, doi:10.3390/ijerph19137992_

Round 1

Reviewer 1 Report

In the reviewed article, the object of research is Military Personnel. Meanwhile, the study is devoted to the study of indicators of physical performance – and it is not at all clear which specialists were actually examined – who it was - computer scientists or sailors. At the same time, in the discussion chapter, the authors note the literature data on the impact of the testing they are studying (p.7 line 271), however, it is not entirely clear from their text which group they examined – materials describing both nutrition and physical activity in the surveyed groups are needed and must be presented in the text. In addition, it is completely unclear in which climatic region at what time the study was conducted – since a number of tests were done indoors, and some outdoors.

The introduction talks a lot about physical performance, but the methodology does not measure the indicator characterizing the maximum oxygen consumption – speaking of performance, the use of objective physiological markers is mandatory – or other objective physiological markers - otherwise the results obtained in the work do not look convincing. The biochemical markers used by the authors look interesting, but their informative value is not always obvious, which the authors themselves state. The paper does not mention the indicators of maximum oxygen consumption or other physiological markers – mandatory for this kind of research and its discussion.

In the work, the authors studied the nutrition of the surveyed groups (p.3 line 114), before the study and after – however, these data are not provided, although in the context of the resensed study, these data are very important and absolutely necessary.  And the same is true regarding the level of physical performance performed by the authors in the framework of the questionnaire (PARQ+) bcgjkmpjdfyyjuj during the study (p.3 line 104).

In addition, in the text - it is unclear and absolutely necessary to correct what the authors mean - in the Introduction (p1 line 39), they write about physical performance, but their tests are actually engaged in special performance, and the emphasis is on tests for anaerobic performance.

Table 1 — the difference in weight between the study groups is 3 kg – and this is quite a lot, especially considering the package of tests used for testing. In the same connection - a test for a vertical jump — it is completely unclear how and how the height of the jump was measured. In addition, is there any data on the time spent in the air, which device was used for measuring or which kind of stability platform – if it was used in this case. In general, the methodological part of the study is written very sparingly and raises a number of questions specifically on the details of measurements and procedures, which is difficult to perceive when reading the results of the study.

The materials and methods section does not describe the Subtraction test, but it is in the results.

In general, the limited set of biochemical markers in the work is unclear, which is unexpected for this kind of research, and what really meant the time of blood sampling for lactate – immediately indicated in the text and where the blood was taken from? 

As can be understood from the text - the test — Stimulated Casualty Evacuation test (p.3 line 141), is the development of the authors, because there is no literary reference to it. It is unclear how valid it is, whether there are correlations with indicators of physical performance. 

In addition, all participants were given the same weight in the trials, but there was a difference in the groups (a tendency) in their weight by 3 kg, which could be significant in the results obtained.

The paper does not provide a justification for the dosages of the drugs used and there is no discussion of this issue.

Table 2 (p5 line 187) is completely unsuccessful – statistical differences of the studied parameters are not shown, although they are discussed in the text. In general, this table needs to be reworked or 2 tables should be formed from it to understand the data obtained by the authors.

Table 3 (p6 line 224) – the title is unsuccessful - the resting level is indicated, but this does not apply to the lactate indicators presented in the table – and is stylistically difficult to perceive.

An important omission of the work is the lack of detailed consideration and description of the relationship between creatinine intake and aerobic performance.

Of particular surprise is the lack of discussion of the most significant phenomenon established in the work – the increase in the testosterone index (p.8 line 305), instead there is a minimal statement of fact and nothing more.

Author Response

We would love to thank the reviewer for providing great comments/insights which absolutely  helped us to improve the quality of the manuscript. 

In the reviewed article, the object of research is Military Personnel. Meanwhile, the study is devoted to the study of indicators of physical performance – and it is not at all clear which specialists were actually examined – who it was - computer scientists or sailors. At the same time, in the discussion chapter, the authors note the literature data on the impact of the testing they are studying (p.7 line 271), however, it is not entirely clear from their text which group they examined – materials describing both nutrition and physical activity in the surveyed groups are needed and must be presented in the text. In addition, it is completely unclear in which climatic region at what time the study was conducted – since a number of tests were done indoors, and some outdoors.

  • The soldiers participated in this study were land forces in operation . The detail has been added to the text.
  • There were no outdoor tests and all procedures were done in an indoor area.
  • The climate was controlled, and this has been added to the text.

The introduction talks a lot about physical performance, but the methodology does not measure the indicator characterizing the maximum oxygen consumption – speaking of performance, the use of objective physiological markers is mandatory – or other objective physiological markers - otherwise the results obtained in the work do not look convincing. The biochemical markers used by the authors look interesting, but their informative value is not always obvious, which the authors themselves state. The paper does not mention the indicators of maximum oxygen consumption or other physiological markers – mandatory for this kind of research and its discussion.

  • Thank you for your comment. While we do agree that determination of one’s VO2max is a strong indicator of endurance performance, the purpose of our study and the physiological rationale of creatine supplementation, in particular, is to augment anaerobic performance. Phosphocreatine is rapidly broken down into Cr and ATP during intense muscular contractions without the use of oxygen. Furthermore, it is well-establish that creatine supplementation has no effect on maximal oxygen consumption (PMID: 34859731). In addition, it is important to note that our study used a number of validated performance measures (1-repetition maximum, vertical jump, anaerobic tests, etc) in addition to a job specific validated test (the simulated casualty evacuation test).

In the work, the authors studied the nutrition of the surveyed groups (p.3 line 114), before the study and after – however, these data are not provided, although in the context of the resensed study, these data are very important and absolutely necessary.  And the same is true regarding the level of physical performance performed by the authors in the framework of the questionnaire (PARQ+) bcgjkmpjdfyyjuj during the study (p.3 line 104).

  • In this study, we did not intend to monitor participants’ diet. Participants were instructed to record the food they consumed in the last day prior to the first testing in order to replicate this diet in the last day prior to post-test. We agree that not controlling for nutrition throughout the entire duration of the study is a limitation, however, this type of methodology is very challenging and expensive and ultimately was not feasible.
  • We are not sure we understand the reviewer’s point regarding to (PARQ+)

In addition, in the text - it is unclear and absolutely necessary to correct what the authors mean - in the Introduction (p1 line 39), they write about physical performance, but their tests are actually engaged in special performance, and the emphasis is on tests for anaerobic performance.

  • The battery of tests consists of power, strength, aerobic, and anerobic test and we do not consider them as special performance. Being successful in army tasks require a wide range of physical fitness including endurance, power, speed, and strength. Most of the tests used in this study were common tests in non-laboratory conditions and their validity is well stablished. Moreover, the last test (SCET), participants were required to perform a task in which they had to challenge their aerobic and anaerobic systems. In previous studies, fitness parameters of military personnel were measured even by running on treadmill. For example, and as previous studies have mentioned, we used 1600 running in order to simulate a rapid approach to battlefield. Previous research has suggested that a prolonged run with sprint is a standard approach to the battlefield (a).
  1. Harman EA, Gutekunst DJ, Frykman PN, Nindl BC, Alemany JA,Mello RP, Sharp MA (2008) Effects of two different eightweek training programs on military physical performance The. J Strength Cond Res 22:524–534
  • Most of our physical performance tests focused on anerobic performance since the nature and concept of B-alanine and creatine is to improve anaerobic performance. Meanwhile, there are studies that reports both B-alanine and creatine are beneficial to memory and brain function.

Table 1 — the difference in weight between the study groups is 3 kg – and this is quite a lot, especially considering the package of tests used for testing. In the same connection - a test for a vertical jump — it is completely unclear how and how the height of the jump was measured. In addition, is there any data on the time spent in the air, which device was used for measuring or which kind of stability platform – if it was used in this case. In general, the methodological part of the study is written very sparingly and raises a number of questions specifically on the details of measurements and procedures, which is difficult to perceive when reading the results of the study.

  • Thank you for your comment, we have revised the methodology section and added further details to the vertical jump test.
  • We agree on an individual level that 3 kg may alter performance, however, it is important to note that this is a group mean with a large amount of variation, as such, there was no statistical difference between groups thus overall, we believe that this group difference is likely not going to make a significant effect.
  • We were unable to use any platforms for testing vertical jump
  • The material and methods section has been improved and written in more details.

The materials and methods section does not describe the Subtraction test, but it is in the results.

  • 7-subtraction test is described in the material and methods section under the subheading of “mathematical processing”. The consistent heading both in material and methods and results section has been used.

In general, the limited set of biochemical markers in the work is unclear, which is unexpected for this kind of research, and what really meant the time of blood sampling for lactate – immediately indicated in the text and where the blood was taken from? 

  • Military personnel are required to train and operate in challenging multi-stressor environments, which can affect hormonal levels, and subsequently results in declining physical performance, cognition, and recovery. There are two main objectives for evaluating anabolic and catabolic conditions in military. First, disruption between anabolic and catabolic hormones may extend recovery times, suppress immune system, and diminish physical and cognitive performance. Military operation or exercise/training induced elevations in cortisol levels can persist for up to 2 weeks following military training (a, b). Similarly, reductions of testosterone might be present up to two weeks post-training (a,b,c). Second, participants in our study performed a set of power, strength, anaerobic, aerobic, and cognitive tests. Being aware of changes in anabolic (testosterone and IGF-1) and catabolic (Cortisol) hormones would help us to interpret whether the possible gains in physical and cognitive performance after supplementation are related to these hormones.
  1. Hamarsland H, Paulsen G, Solberg PA, Slaathaug OG, Raastad T. Depressed Physical Performance Outlasts Hormonal Disturbances after Military Training. Medicine and science in sports and exercise. 2018.
  2. Szivak TK, Lee EC, Saenz C, Flanagan SD, Focht BC, Volek JS, et al. Adrenal stress and physical performance during military survival training. Aerospace medicine and human performance. 2018; 89 (2):99–107.
  3. Ojanen T, Kyro¨ la¨inen H, Igendia M, Ha¨kkinen K. Effect of prolonged military field training on neuromuscular and hormonal responses and shooting performance in warfighters. Mil Med. 2018.
  • According to previous studies, blood lactate reaches peak concentration usually up to the seventh minute after exercise (a, b, and c). we measured blood lactate between 1 to 3 minutes post exercise while participants were passively resting in a comfortable position. Lactate was measured after the series of intense physical activities in order to see if supplementation affect post-exercise lactate concentration.
  • Blood lactate was measured according to the manufacturer instructions (Lactate Scout 4, Germany) from the tip of index finger. Details were added to the text.
  1. Beneke R, Hütler M, JungMet al. Modeling the blood lactate kinetics at maximal short-term exercise conditions in children, adolescents, and adults. J Appl Physiol 2005; 99:499–504.
  2. Gass GC, Rogers S, Mitchell R. Blood lactate concentration following maximum exercise in trained subjects. Br J Sports Med 1981; 15:172–176.
  3. Machado, Fabiana A., Ana Claudia P. Kravchychyn, Cecilia S. Peserico, Danilo F. da Silva, and Paulo V. Mezzaroba. "Effect of stage duration on maximal heart rate and post-exercise blood lactate concentration during incremental treadmill tests." Journal of science and medicine in sport 16, no. 3 (2013): 276-280.

As can be understood from the text - the test — Stimulated Casualty Evacuation test (p.3 line 141), is the development of the authors, because there is no literary reference to it. It is unclear how valid it is, whether there are correlations with indicators of physical performance. 

  • The simulated casualty evacuation test (SCET) in this paper is a modified test from previous research. All of the previous studies have used running/sprinting and carrying casualty. However, they are different in terms of the weight of casualty, and the distance that participants ran.
  1. Poser, Whitney M., Kara A. Trautman, Nathan D. Dicks, Bryan K. Christensen, Katie J. Lyman, and Kyle J. Hackney. "Simulated casualty evacuation performance is augmented by deadlift peak force." Military medicine184, no. 9-10 (2019): e406-e411.
  2. Varanoske, Alyssa N., Adam J. Wells, Gregory J. Kozlowski, Yftach Gepner, Cheyanne L. Frosti, David Boffey, Nicholas A. Coker, Idan Harat, and Jay R. Hoffman. "Effects of β‐alanine supplementation on physical performance, cognition, endocrine function, and inflammation during a 24 h simulated military operation." Physiological Reports6, no. 24 (2018): e13938.
  3. Hoffman, Jay R., Geva Landau, Jeffrey R. Stout, Mattan W. Hoffman, Nurit Shavit, Philip Rosen, Daniel S. Moran et al. "β-Alanine ingestion increases muscle carnosine content and combat specific performance in soldiers." Amino acids 47, no. 3 (2015): 627-636.

In addition, all participants were given the same weight in the trials, but there was a difference in the groups (a tendency) in their weight by 3 kg, which could be significant in the results obtained.

  • We agree that there is a slight difference, however, this is not statistically different between groups (pre-test weight p=0.404, post-test weight, p=0.402). in addition, the weight of the rifle and the weight in casualty was not modified among participants with different weight since there is no modification in battlefield for different individuals.

The paper does not provide a justification for the dosages of the drugs used and there is no discussion of this issue.

  • The doses used were based on ISSN positions stands: Beta-Alanine: https://pubmed.ncbi.nlm.nih.gov/26175657/ and Creatine: https://pubmed.ncbi.nlm.nih.gov/28615996/. The dosing strategies used in our study are commonly used and recommended by sport nutrition organizations.

Table 2 (p5 line 187) is completely unsuccessful – statistical differences of the studied parameters are not shown, although they are discussed in the text. In general, this table needs to be reworked or 2 tables should be formed from it to understand the data obtained by the authors.

  • Thank you for the comments. The table 2 has been revised for clarity,

Table 3 (p6 line 224) – the title is unsuccessful - the resting level is indicated, but this does not apply to the lactate indicators presented in the table – and is stylistically difficult to perceive.

  • The title of table has been modified.

An important omission of the work is the lack of detailed consideration and description of the relationship between creatinine intake and aerobic performance.

  • Thanks for the comment. More relevant information has been added to the text. It is important to note that creatine does not appear to enhance aerobic performance, but may be of benefit during periods of higher intensity during endurance/aerobic exercise (e.g., https://pubmed.ncbi.nlm.nih.gov/28806275/).

Of particular surprise is the lack of discussion of the most significant phenomenon established in the work – the increase in the testosterone index (p.8 line 305), instead there is a minimal statement of fact and nothing more.

  • Thanks for the comment. We have provided further discussion of these findings.

Reviewer 2 Report

The manuscript is prepared relatively correctly.  However, there are a few critical issues that should be addressed before possible publication acceptation.

In this manuscript, the authors attempted to investigate the impact of creatine and beta-alanine co-supplementation on physiological and biochemical indices in physically active males. This is a topic that interests scientists, coaches and athletes, and deserves further research. Due to the more frequent occurrence of supplements in athletes, this problem is very important. However, in its present form, many shortcomings preclude this manuscript from being published in the journal. My detailed comments are as follows:

Lines 20-21 Avoid using gr (grams). More proper is g. Besides, in the rest of the manuscript you will use gr once, another time g (mg) [line 110]

Line 51 – upper index H+

Line 62-64 connect these to sentences of carnosine to one

Line 64 – “A growing body literature” change to “A growing body of literature”

Line 106 – Authors should add an approval of the relevant bioethics committee

Lines 101-117 – You actually didn't describe the form of supplements creatine (monohydrate or other?). You have to add more information about used forms of both supplements (manufacturer including).

Lines 112-113 - Did the participants took their dose by any part of the day. Did they took one portion of supplement daily or they had guidelines for a specific divide daily dose of the supplement (as you write for breakfast, lunch and dinner)?

Line 161 – At what time of the day was the blood collected from the individual participants? Remember that steroid hormones, and cortisol in particular, fluctuate widely throughout the day (daily cycle).

Lines 162-164 – What kind of blood was collected to lactate determination? Did the rest values of lactate were measured?

Line 164 – write correct - Lactate Scout (LS, SensLab GmbH, Germany)

Line 167 – write more specifically (for example – Cat. No.)

Line 168 – Did the sample size was calculated?

Line 178 (table 1) – … demographic characteristic is incorrect definition! 

Line 223 – add the description “post-exercise lactate”

Table 3 – use on standard of units decryption. L or l?

Table 3 – I propose to recalculate the hormones concentrations to /L. These are more common used and presented in research literature.

Table 3 – use micro (m) symbol not u to cortisol concentration

Lines 188 – 245 – The description of results (P values) are not clear. If table P values show the differences between baseline, so what does mean “a significant difference between BA+Cr and BA+PL groups”? I suppose that means between values after supplementation protocol but it is undescribed.

Line 263 – what do you mean “speed”?

Line 290 – upper index H+

Lines 319-324 – Probably is to late to change the analyzed parameters, but in future research I propose to measure (analyze) concentrations of BDNF or/and serotonin or/and dopamine. In my opinion they will be more specific to cognitive skills measurement.

Additionally:

  1. check the manuscript for double spaces or no spaces. Unfortunately, in many places this creates an impression of carelessness.
  2. check carefully the references format characterized to Journal. Some times you write the Journal – capital letters (Journal of the International Society of Sports Nutrition) other time a lowercase letter (Journal of applied physiology).

Author Response

We would love to thank the reviewer for their constructive and helpful comments/suggestions on our manuscript. 

The manuscript is prepared relatively correctly.  However, there are a few critical issues that should be addressed before possible publication acceptation.

In this manuscript, the authors attempted to investigate the impact of creatine and beta-alanine co-supplementation on physiological and biochemical indices in physically active males. This is a topic that interests scientists, coaches and athletes, and deserves further research. Due to the more frequent occurrence of supplements in athletes, this problem is very important. However, in its present form, many shortcomings preclude this manuscript from being published in the journal. My detailed comments are as follows:

Lines 20-21 Avoid using gr (grams). More proper is g. Besides, in the rest of the manuscript you will use gr once, another time g (mg) [line 110]

  • Thanks for the comment. We have modified the manuscript accordingly.

Line 51 – upper index H+

  • All of the H+ have changed to H+.

Line 62-64 connect these to sentences of carnosine to one

  • These sentences have been combined to one sentence.

Line 64 – “A growing body literature” change to “A growing body of literature”

  • It has been corrected as suggested.

Line 106 – Authors should add an approval of the relevant bioethics committee

  • It was added under the heading ‘Institutional Review Board Statement’, page 8.

Lines 101-117 – You actually didn't describe the form of supplements creatine (monohydrate or other?). You have to add more information about used forms of both supplements (manufacturer including).

  • More detail has been added.

Lines 112-113 - Did the participants took their dose by any part of the day. Did they took one portion of supplement daily or they had guidelines for a specific divide daily dose of the supplement (as you write for breakfast, lunch and dinner)?

  • For beta-alanine capsules, participants were asked to take 8 capsules (6.4g) with their meals. There was no firm instruction, however, they were recommended to take 3 capsules with breakfast, 3 with lunch, and 2 with their dinner.
  • For creatine, participants were asked to divide the creatine powder to 3 portions and disolve in water and drink with their meal.

Line 161 – At what time of the day was the blood collected from the individual participants? Remember that steroid hormones, and cortisol in particular, fluctuate widely throughout the day (daily cycle).

  • The resting blood samples were collected following a 12h-fast in the morning. This information has been added to the text.

Lines 162-164 – What kind of blood was collected to lactate determination? Did the rest values of lactate were measured?

  • The capillary blood from the tip of index finger was used.
  • Resting values of lactate was not measured.

Line 164 – write correct - Lactate Scout (LS, SensLab GmbH, Germany)

  • It has been modified.

Line 167 – write more specifically (for example – Cat. No.)

  • The Cat number was inserted.

Line 168 – Did the sample size was calculated?

  • The sample size was not determined based on an a priori calculation. It was based on previous published similar research and feasibility.

Line 178 (table 1) – … demographic characteristic is incorrect definition! 

  • It has been modified

Line 223 – add the description “post-exercise lactate”

  • Thanks for the comment. It has been modified.

Table 3 – use on standard of units decryption. L or l?

  • It has been modified.

Table 3 – I propose to recalculate the hormones concentrations to /L. These are more common used and presented in research literature.

  • The values have been recalculated based on concentration per liter. .

Table 3 – use micro (m) symbol not u to cortisol concentration

  • It has been modified.

Lines 188 – 245 – The description of results (P values) are not clear. If table P values show the differences between baseline, so what does mean “a significant difference between BA+Cr and BA+PL groups”? I suppose that means between values after supplementation protocol but it is undescribed.

  • The statement has been clarified. The statement “a significant difference between BA+Cr and BA+PL groups” refers to post-supplementation values.

Line 263 – what do you mean “speed”?

Line 290 – upper index H+

  • We have modified accordingly.

Lines 319-324 – Probably is to late to change the analyzed parameters, but in future research I propose to measure (analyze) concentrations of BDNF or/and serotonin or/and dopamine. In my opinion they will be more specific to cognitive skills measurement.

  • This is a great suggestion and it could have been helpful for interpreting possible cognitive gains. Unfortunately, we were unable to measure BDNF/Serotonin/Dopamine and this was considered as the limitation of our study.

 Additionally:

  1. check the manuscript for double spaces or no spaces. Unfortunately, in many places this creates an impression of carelessness.
  • It has been checked.
  1. check carefully the references format characterized to Journal. Sometimes you write the Journal – capital letters (Journal of the International Society of Sports Nutrition) other time a lowercase letter (Journal of applied physiology).
  • We have checked the references carefully.

Reviewer 3 Report

A fairly clearly written paper, but I have some concern whether your design allows you to adequately address the synergistic effects of beta-alanine and creatine given the lack of a placebo group and creatine only group.

Points to consider:

Introduction - It would be informative to include a sentence on why/how creatine might enhance cognitive performance.

I think the paragraph reporting previous studies of co-ingestion of beta-alanine and creatine needs revising. For example, you report that Zoeller et al. only observed within group changes for beta-alanine and creaitne combined, but their paper reports within group improvements in TTE and power at VT from creatine alone and power output at LT for Beta-alanine alone. Please check other studies in this paragrpah have been accurately described.

Methods: How did you decide on your sample size of 20? Did you conduct a sample size calculation?

Why did you not include a placebo group or a creatine only group? Without these I am not convinced you can conclude whether beta-alanine has any effect or whether creatine has a synergistic effect.

Why did you decide to supplement creatine for 7 days?

Biochemical analysis section 2.4: 

I think this section would be clearer if this sentence was moved to the end of the paragraph.

"At the end of the last physical test (SCET), both in pre-test and post-test, blood lactate was immediately measured using a lactate analyzer 4 (Scout, Germany)"

Statistical analysis - If you used ANCOVA, what covariates were included in your model?

Results:

Tables 2 & 3: It seems a little confusing to present the p values for baseline group comparisons in this results table rather than between group treatment effects. 

Discussion:

I think it is difficult to interpret within groups changes in a RCT if there are no between group differences and no placebo group although I guess you have acknowledged this.

Author Response

We would love to thank the reviewer for their constructive and helpful comments/suggestions on our manuscript. 

Comments and Suggestions for Authors

A fairly clearly written paper, but I have some concern whether your design allows you to adequately address the synergistic effects of beta-alanine and creatine given the lack of a placebo group and creatine only group.

Points to consider:

Introduction - It would be informative to include a sentence on why/how creatine might enhance cognitive performance.

  • Thanks for the comment. More information has been added.

I think the paragraph reporting previous studies of co-ingestion of beta-alanine and creatine needs revising. For example, you report that Zoeller et al. only observed within group changes for beta-alanine and creaitne combined, but their paper reports within group improvements in TTE and power at VT from creatine alone and power output at LT for Beta-alanine alone. Please check other studies in this paragrpah have been accurately described.

Methods: How did you decide on your sample size of 20? Did you conduct a sample size calculation?

  • The sample size was based on feasibility. Furthermore, similar supplement studies used a similar sample size. A sample size calculation was not done a priori and we have mentioned this as a limitation.

Why did you not include a placebo group or a creatine only group? Without these I am not convinced you can conclude whether beta-alanine has any effect or whether creatine has a synergistic effect.

  • We agree, the design only allows us to determine if creatine in addition to beta-alanine is of further benefit than beta-alanine alone. We have carefully revised the manuscript for clarity.

Why did you decide to supplement creatine for 7 days?

  • This strategy has been shown to saturate muscles (Hultman et al. 1996: PMID: 8828669).

Biochemical analysis section 2.4: 

I think this section would be clearer if this sentence was moved to the end of the paragraph.

"At the end of the last physical test (SCET), both in pre-test and post-test, blood lactate was immediately measured using a lactate analyzer 4 (Scout, Germany)"

  • It has been relocated.

Statistical analysis - If you used ANCOVA, what covariates were included in your model?

  • We used baseline values as the co-variate.

Results:

Tables 2 & 3: It seems a little confusing to present the p values for baseline group comparisons in this results table rather than between group treatment effects. 

  • Thanks for the comment. The table has been modified.

Discussion:

I think it is difficult to interpret within groups changes in a RCT if there are no between group differences and no placebo group although I guess you have acknowledged this.

  • We agree that this design has limitations, however, we feel that this study adds to the growing body of literature examining supplement co-ingestion.

Round 2

Reviewer 1 Report

The authors have significantly revised the description of the methodological part of the study and now it has become more clear what methods and approaches were used to evaluate the intake of dietary supplements on the functional state of the examined servicemen.

At the same time , in fact , from the data obtained by the authors , it can be noted as a novelty:

1. an increase in testosterone content (which is well known in the literature) after taking BA – but the authors describe it only under the influence of BA+PL, but not in the BA+Cr group (table on P5 L244) – for some reason this table is again designated as Table 1 – whereas the first table in the text is on P5 Line201);

2. The increase in vertiucal Jump (table2) established by the authors in the BA+Cr group, however, was not measured instrumentally, but in centimeters of the increase in the height of the touch in the jump when testing was completed – after all, the method has a subjective component. It can be noted that in the second study, the jump height indicators during the second test in the BA+Cr group only reached the levels in the BA+PL group – although the growth of the examined groups did not differ significantly, and in general, in the first group as a whole, the soldiers were slightly taller (table1 - P5 Line201);

In addition, the authors write P8 Line316-317 In support of this hypothesis, the within-group comparison showed a significant increase in chest and leg strength in the BA+Cr group, while no change was found in the BA+PL group – however, this does not follow from the materials of table 2 P5 below Line207 – or the authors could not or forgot to specify – which in fact does not change the situation.

Author Response

We would love to thank the reviewer for their helpful and constructive comments on our manuscript which helped us to improve the quality/clarity of the manuscript. We have addressed your comments as follows.

Regarding this comment, testosterone levels increased from 5.46 ng/l to 5.92 ng/l in the BA+CR group while it was from 5.61 ng/l to 5.64 ng/l indicating a significant difference between the groups. We realize that there was an error in the process of changing the units as one of the reviewers has requested us to do so.  Therefore, we have reported that there is a significant increase in testosterone levels in BA+CR compared to BA+PR, while there is still a slight increase in testosterone levels in the BA+PR within-group which was not significant.

We agree with the reviewer with respect to the non-instrumental method of measuring vertical jump; however, we have used the same standing reach height which was measured in the pre-test time point in the subsequent testing to minimize errors.

Table 2 only indicates between group differences rather than within group differences. In line 216-225 page 6, we have mentioned the within group differences which is in line with this statement.